

# Assembly and comparative analysis of complete mitochondrial genome sequence of an economic plant *Salix suchowensis*

Ning Ye[1], Xuelin Wang[1], Juan Li[2], Changwei Bi[3], Yiqing Xu[1], Dongyang Wu[4] and Qiaolin Ye[1]

[1] College of Information Science and Technology, Nanjing Forestry University, Nanjing, Jiangsu, China
[2] School of Electrical and Automatic Engineering, Nanjing Normal University, Nanjing, Jiangsu, China
[3] School of Biological Science and Medical Engineering, Southeast University, Nanjing, Jiangsu, China
[4] College of Forest Resources and Environment, Nanjing Forestry University, Nanjing, Jiangsu, China

## ABSTRACT

Willow is a widely used dioecious woody plant of *Salicaceae* family in China. Due to their high biomass yields, willows are promising sources for bioenergy crops. In this study, we assembled the complete mitochondrial (mt) genome sequence of *S. suchowensis* with the length of 644,437 bp using Roche-454 GS FLX Titanium sequencing technologies. Base composition of the *S. suchowensis* mt genome is A (27.43%), T (27.59%), C (22.34%), and G (22.64%), which shows a prevalent GC content with that of other angiosperms. This long circular mt genome encodes 58 unique genes (32 protein-coding genes, 23 tRNA genes and 3 rRNA genes), and 9 of the 32 protein-coding genes contain 17 introns. Through the phylogenetic analysis of 35 species based on 23 protein-coding genes, it is supported that *Salix* as a sister to *Populus*. With the detailed phylogenetic information and the identification of phylogenetic position, some ribosomal protein genes and succinate dehydrogenase genes are found usually lost during evolution. As a native shrub willow species, this worthwhile research of *S. suchowensis* mt genome will provide more desirable information for better understanding the genomic breeding and missing pieces of sex determination evolution in the future.

## INTRODUCTION

Salicaceae, a family of dioecious catkin-bearing woody plants mainly including willows (*Salix*), poplars (*Populus*) and *Chosenia*, are well known for their worldwide diverse uses (*Dai et al., 2014*). Most willow species with small, short or exceptional stems flower early during their lives and thus differ from poplars in their living habits and histories (*Stettler et al., 1996*). As a widespread bush willow species of Salicaceae family in China, *S. suchowensis* could reach sexual maturity in one year to propagate.

Mitochondria are organelles, whose main function is to convert the energy from biomass energy into chemical energy in living cells. The origin of plastids and mitochondria are through the endosymbiotic acquisition of formerly free-living bacteria, which can explain the presence of own genomes (*Greiner & Bock, 2013*; *Keeling, 2010*; *Sagan, 1967*). After more than a billion years of concerted subsequent evolution to optimize the compartmentalized

Corresponding author
Ning Ye, yening@njfu.edu.cn

genetic material expression, the three genomes of plant cells (nucleus, plastid and mitochondrion) have undergone dramatic structural changes (*Greiner & Bock, 2013*). For most seed plants, nuclear hereditary information is inherited biparentally, while both chloroplasts (cp) and mitochondria are derived from maternal genes (*Birky, 1995*). The mt genome plays a significant role in plant productivity and development (*Yasunari et al., 2005*). Previous studies showed that plant mt genomes with abundant genes were large and complex compared to other non-plant eukaryotes (*Li et al., 2009b*; *Yang et al., 2011*). During the evolution of plant mt genomes, land plants probably have obtained some new mechanisms to assist more frequent gene exchanges between mt and nuclear genomes, as well as mt and cp genomes (*Fujii et al., 2010*). It is also suggested that seed plant mt genomes are distinct for their very low mutation rate generally (*Palmer & Herbon, 1988*; *Wolfe, Li & Sharp, 1987*), dynamic structure (*Lonsdale et al., 1988*), repeatedly uptake of foreign DNA by intracellular and horizontal gene transfer (*Richardson & Palmer, 2007*; *Stern & Lonsdale, 1982*), relatively high incidence of *trans*-splicing of coding sequences and RNA editing (*Knoop, 2004*; *Schuster, Hiesel & Brennicke, 1990*), and their extremely large and highly variable sizes (*Hsu & Mullin, 1989*; *Ward, Anderson & Bendich, 1981*).

An increasing number of organelle genomes have been completed in the past several years. Until now, there are 255 complete mt genomes having been deposited in GenBank Organelle Genome Resources. The plant mt genomes vary considerably in length (varying from 200 to 700 kb mostly), gene order and gene content (*Richardson et al., 2013*). The largest published land plants mt genome of *Silene conica* is 11.3 Mb in length (*Sloan et al., 2012*) and the smallest of known land plants mt genome is about 66 kb (*Viscum scurruloideum*) in length (*Skippington et al., 2015*). The gene content of land plants also varies considerably, mainly ranging from 32 to 67 genes, and some genes like NADH dehydrogenase subunit are encoded in discontinuous fragments that require *trans*-splicing event (*Hsu & Mullin, 1989*; *Ward, Anderson & Bendich, 1981*). With the emergence of next-generation sequences, new approaches for high-throughput, timesaving and low-cost genome sequencing will be applied gradually. *Populus trichocarpa* was a well-studied model tree in many other different types of research and it was the first sequenced wood plant in the family of Salicaceae (*Tuskan et al., 2006*). In this study, we determined the complete mt genome sequence of *S. suchowensis*. The complete cp genome of this specie has already been reported (*Wu, 2016*), but its mt genome has not been determined. We also analyze the genomic and structural features of *S. suchowensis* mt genome and make a comparison with other angiosperms (and gymnosperms at the same time, which will provide information for the better understanding of mt genome evolution in land plants. Further and more importantly, this study could bring more desirable information for better understanding the genetic transformation, genomic breeding (*Wu, 2016*) and missing pieces of sex determination evolution in the future (*Kersten et al., 2016*).

## MATERIALS AND METHODS

### Genome sequencing preparation

To construct, amplify and sequence the original genome sequencing libraries, we used Roche/454 and Illumina/HiSeq-2000 sequencing technologies according to the

manufacture's kits and protocols. The *S. suchowensis* mt genome sequencing was conducted with a combined approach using Roche/454 and Illumina/HiSeq-2000 sequencing technologies at Nanjing Forestry University following the manufacturer's protocols (*Dai et al., 2014*; *Liu et al., 2013*).

## Genome assembly and annotation

To generate a scaffold-level and gap-free mitochondrial genome of *S. suchowensis*, we used the 454 GS FLX Titanium platform (454 Life Sciences) in this research. As shown in Table S1, we got 1,240,387 high-quality reads with a total length of 702,204,081 bp. The average read length is 567 bp, and the longest read length is 1,201 bp. Since the original sequence reads were a mixture of DNA with the nucleus and other organelles like chloroplast, basing on 255 known reference mt genomes download from NCBI, we used BLASTN to isolate mitochondrial reads from the total reads (*Buhler et al., 2007*). Newbler 3.0 (454-Roche) with default parameters was used to assemble the *S. suchowensis* mt genome sequences. After this process, we obtained 83,984 contigs, and the longest contig length is 349,730 bp. Contigs with long length and high reads depth were separated with shorter ones (*Zhang et al., 2011*). We filtered contigs with read depth between 15x to 50x out, which may contain essential mitochondrial genes. According to the manual of the assembly software Newbler (*Zhang et al., 2011*), contigs are constructed and almost no overlaps among these connected contigs. Therefore, we used Perl scripts and Newbler 3.0 (454-Roche) generated file "454AllContigGraph.txt" to determine the contigs connections. According to read depths of the contigs, false links and some wrong forks were removed manually in the meanwhile, and we finally obtained 13 contigs to construct the draft connection graph for assembling the mt genome. As shown in supplementary Table S2 and Fig. S1, three contigs (Contig30222, Contig34550 and Contig55858) resemble cp-derived sequences with their high coverages. However, with the connection in file '454ContigGraph.txt', these three contigs are essential for assembling the complete mt genome, suggesting that these three contigs may be derived from the *S. suchowensis* cp genome. Additionally, 3 of the 13 selected contigs (Contig00022, Contig00027 and Contig55858) were assembled into the complete mt genome twice, indicating that these three contigs are repeat sequences in *S. suchowensis* mt genomes. After that, we used the method described by *Ji et al. (2013)* to fill the gaps of contigs up : first, we mapped mitochondrial reads onto both ends (3–60 bp) of the assembled contigs and then extended the contigs by joining the reads, which were partly overlapped (Identity $\geq$ 95%; $e$-value $\leq 1e-30$) with the contigs.

In order to get a more accurate genome, we used 64 Illumina (Illumina Genome Analyzer II) runs (Accession: SRX1561932, 51.2 Gb) generated by a standard Solexa protocol to validate the genome assembly. Using BWA (*Li & Durbin, 2009*) and SAMtools (*Li et al., 2009a*), we mapped the Illumina sequencing data onto the draft mt genome assembled by Roche 454 sequencing date, and then we filtered the mapped reads with local Perl scripts. After reassembling these mapped reads with Newbler software, a total of 214 contigs (longest length: 16,445 bp, average length: 2,271 bp, N50: 3,254 bp) with total length 486,008 bp were generated. Next, we remapped the 214 contigs onto the draft mt genome in MacVector, and corrected 168 sequencing mistakes generated by 454 GS FLX Titanium

platform (especially in A/T enriched regions). Finally, the complete mt genome sequence of *S. suchowensis* was finished.

The Web-based tool Public MITOFY Analysis was used to identify genes (*Alverson et al., 2010*), accompanying with synonymous and none-synonymous SNPs. We also revised the start and stop codons of genes by learning similar genes, which has been manually checked by MITOFY firstly from other mt genomes sequenced. All transfer RNA genes were confirmed by using tRNAscan-SE with default settings (*Schattner, Brooks & Lowe, 2005*). The circular mt genome map was drawn through the OGDRAW online program (*Lohse, Drechsel & Bock, 2007*).The final mt genome of *S. suchowensis* has been deposited to GenBank (Accession number: NC_029317.1)

## Analysis of repeat structure and sequence

Simple sequence repeats (SSRs) of the *S. suchowensis* mt genome were discovered using Perl script MISA (http://pgrc.ipkgatersleben.de/misa/) (*Thiel et al., 2003*), with the size of one to six nucleotides and thresholds of eight, four, four, three, three, and three, separately. Tandem repeats were analyzed with the aid of Tandem Repeats Finder Program v4.04 (*Benson, 1999*) using default parameters. Additionally, AB-blast was utilized to identify and locate disperse repeats with the following parameters: $M = 1, N = -3, Q = 3, R = 3$, kap, span, $B = 1 \times 10^9$ and $W = 7$, which have been proved effectively in other mt genomes (*Alverson et al., 2010*; *Alverson et al., 2011*; *Bi et al., 2016*). The BLAST hits with *e*-value $\leq 1$ and identity $\geq 80\%$ were considered as disperse repeats. All of the repeats sequence identified with the above-mentioned programs were manually checked to filter excessive results.

## Analysis of RNA editing and substitution rate

The online program Predictive RNA Editor for Plants (PREP) suite (http://prep.unl.edu/) (*Mower, 2009*) was adopted to identify the possible RNA editing sites in the protein-coding genes of *S. suchowensis* mt genome. For the sake of an accurate prediction, the cutoff value was set as 0.2. The protein-coding genes from other plant mt genomes were utilized as references for revealing the RNA editing sites in the *S. suchowensis* mt genome.

In order to analyze the synonymous (Ks) and nonsynonymous (Ka) substitution rates of the protein-coding genes in *S. suchowensis* and other higher plant mt genomes, *Gossypium raimondii* was selected as reference. Protein sequences of the corresponding protein-coding genes in *S. suchowensis* and *G. raimondii* mt genomes were extracted and then aligned separately using ClustalW (*Thompson, Gibson & Higgins, 2002*). The Ks and Ka substitution rates of each protein-coding gene were finally estimated in DnaSP v5.10 with default settings (*Librado & Rozas, 2009*).

## DNA transfer from chloroplast to mitochondrion

The migration of DNA segments between cp and mt genomes has been identified unidirectional from cp to mt genomes in most higher plants (*Chang et al., 2013*), only a few examples of DNA transfer from mt to cp genome (*Smith, 2014*). The cp genome of *S. suchowensis* (NC_026462) was downloaded from the NCBI Organelle Genome Resources database. DNA segments transferred from *S. suchowensis* cp to mt genomes were identified

by NCBI-BLASTn with stringent parameters: -evalue $1e{-}6$, -word_size 9, -gapopen 5, -gapextend 2, -reward 2, -penalty -3, and -dust no. In order to identify cp-derived sequences and detect conserved mt sequences, the *S. suchowensis* cp genome was searched against a database composed of 25 representative seed plant cp genomes using NCBI-BLASTn with the same stringent parameters. Additionally, protein-coding and tRNA genes transferred from cp to mt genomes were also identified by NCBI-BLASTn, and the results with identity $\geq 90\%$, *e*-value $\leq 1e{-}10$ and coverage $\geq 90\%$ were selected as candidates.

### Phylogenetic analysis

In order to acquire the phylogenetic position of *S. suchowensis*, 35 plant mt genomes (*Ajuga reptans, Asclepias syriaca, Arabidopsis thaliana, Batis maritima, Brassica napus, Butomus umbellatus, Beta vulgaris, Capsicum annuum, Citrullus lanatus, Carica papaya, Cucurbita pepo, Cucumis sativus, Cycas taitungensis, Daucus carota, Ginkgo biloba, Glycine max, Gossypium hirsutum, Gossypium raimondii, Lotus japonicas, Malus domestica, Marchantia pinnata, Medicago truncatula, Nicotiana tabacum, Oryza sativa, Phoenix dactylifera, Populus tremula, Raphanus sativus, Rhazya stricta, Sorghum bicolor, Salvia miltiorrhiza, Salix suchowensis, Tripsacum dactyloides, Vigna angularis, Vitis vinifera* and *Zea mays*) were downloaded from the NCBI Organelle Genome Resources database (http://www.ncbi.nlm.nih.gov/genome/organelle/). Among these species, not only the complete mt genome sequences of these species for analysis were available in NCBI, but also, they were placed clearly in taxonomy and were widely used. Phylogenetic analyses were performed on 23 conserved protein-coding genes (*atp1, atp4, atp6, atp8, atp9, ccmB, ccmC, ccmFc, ccmFn, cob, cox1, cox2, cox3, matR, nad1, nad2, nad3, nad4, nad4L, nad5, nad6, nad7* and *nad9*), which were extracted from these 35 plant mt genomes by local Perl scripts. These conserved genes were then aligned using Muscle (*Edgar, 2004*) implemented in MEGA 6.0 (*Tamura et al., 2013*), and the alignment was modified manually to eliminate gaps and missing data. Finally, a Neighbor-joining (NJ) tree was constructed in MEGA 6.0 with the Poisson model, and the rates among sites were set as Uniform rates. The bootstrap consensus tree was inferred from 1000 replications. *C. taitungensis* and *G. biloba* were designated as the outgroup.

## RESULTS AND DISCUSSION

### Genomic features of the *S. suchowensis* mt genome

The complete mitochondrial genome of *S. suchowensis* is 644,437 bp in length (GenBank accession number: NC_029317.1) with a typical circular structure of land plant genomes. The overall base composition of the entire genome is as follows: A: 27.43%, T: 27.59%, G: 22.64%, C: 22.34%, and the GC content is 44.98%. The positions of all the genes identified in the *S. suchowensis* mt genome and functional categorization of these genes are presented in Fig. 1. A total of 58 unique genes were identified in the *S. suchowensis* mt genome, including 32 protein-coding genes, 23 tRNA genes and 3 rRNA genes (Table 1). Interestingly, the mt genome contains two copies of *rps7* gene.

As shown in Table 1, most of the protein-coding genes use ATG as their start codon, except *rpl16* and *mttB*, which have undetermined start codons. Two genes (*rpl16* and
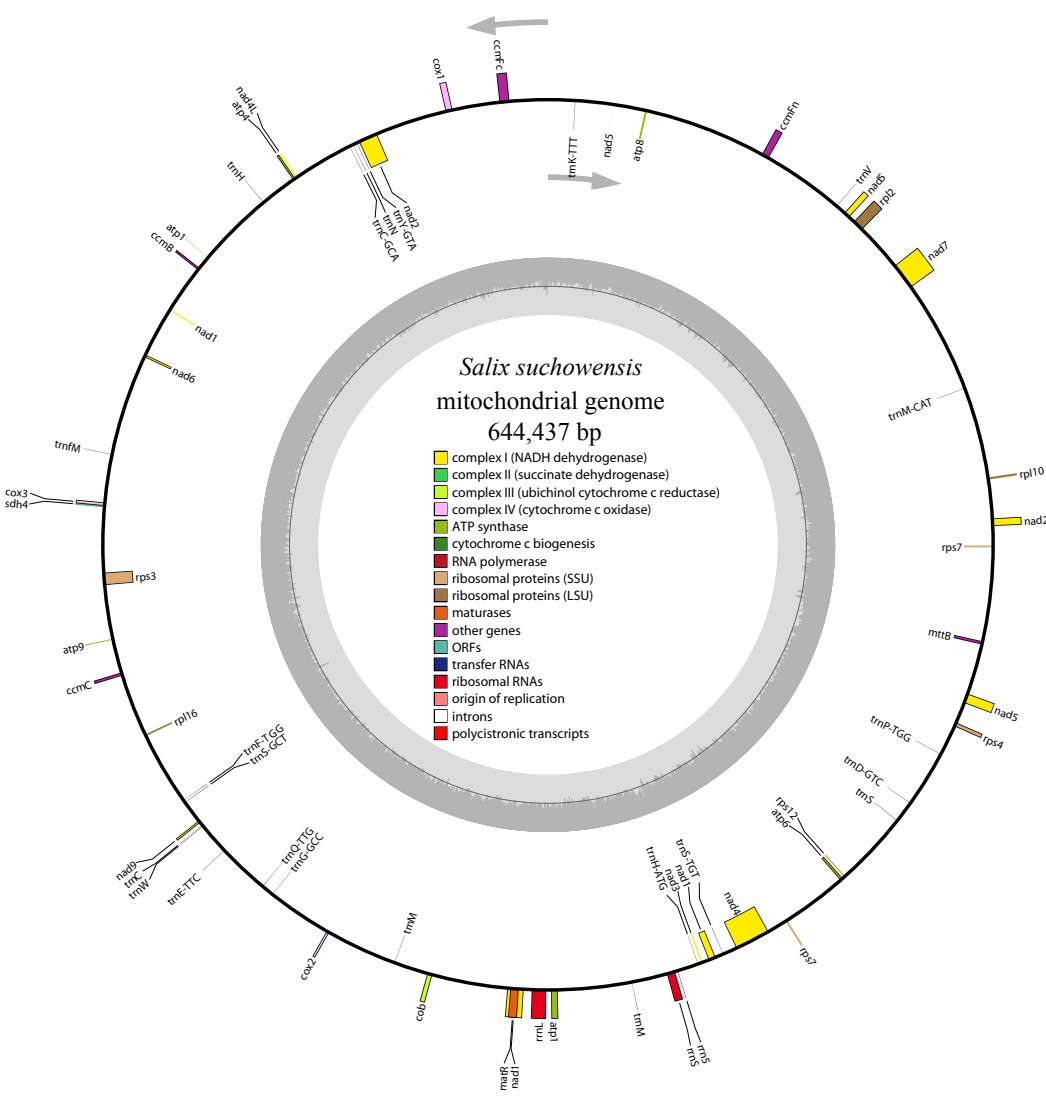

**Figure 1** **The circular mitochondrial genome of *S. suchowensis*.** Genomic features on transcriptionally clockwise and counter-clockwise strands are drawn on the inside and outside of the circle, respectively. Genes belonging to different functional groups are color-coded. GC content is represented on the inner circle by the dark gray plot.

*mttB*) have been reported to have undetermined start codons in many higher plant mt genomes (*Bi et al., 2016*; *Handa, 2003*; *Oda et al., 1992*). The usages of the stop codons in *S. suchowensis* mt protein-coding genes are TAA (16 genes; *atp6*, *cox1*, *cox2*, *nad1*, *nad2*, *nad3*, *nad4L*, *nad5*, *nad9*, *rpl2*, *rpl10*, *rpl16*, *rps3*, *rps4*, *rps7* and *sdh4*), TAG (8 genes; *atp4*, *atp8*, *atp9*, *ccmFc*, *mttB*, *matR*, *nad6* and *nad7*) and TGA (8 genes; *atp1*, *ccmB*, *ccmC*, *ccmFn*, *cob*, *cox3*, *nad4* and *rps1*), suggesting that there is no C to U RNA editing event happened in the stop codons. Furthermore, with the purpose of exhibiting the details of the *S. suchowensis* mt genome better, a GBrowse (Fig. S2) was built at http://bio.njfu.edu.cn/gb2/gbrowse/Salix_Suchowensis_mt/.

**Table 1  Gene profile and organization of the *Salix Suchowensis* mitogenome.**

| Group of genes | Gene name | Length | Start codon | Stop codon | Amino acid |
|---|---|---|---|---|---|
| | *atp1* | 1,596 | ATG | TGA | 531 |
| | *atp4* | 597 | ATG | TAG | 198 |
| ATP synthase | *atp6* | 714 | ATG | TAA | 237 |
| | *atp8* | 474 | ATG | TAG | 157 |
| | *atp9* | 225 | ATG | TAG | 75 |
| | *ccmB* | 615 | ATG | TGA | 204 |
| | *ccmC* | 753 | ATG | TGA | 251 |
| Cytochrome c biogenesis | *ccmFc*[a] | 1,362 | ATG | TAG | 453 |
| | *ccmFn* | 1,725 | ATG | TGA | 575 |
| Ubichinol cytochrome c reductase | *cob* | 1,182 | ATG | TGA | 393 |
| | *cox1* | 1,584 | ATG | TAA | 527 |
| Cytochrome c oxidase | *cox2* | 675 | ATG | TAA | 224 |
| | *cox3* | 798 | ATG | TGA | 266 |
| Maturases | *matR* | 1,944 | ATG | TAG | 647 |
| Transport membrane protein | *mttB* | 786 | ND | TAG | 261 |
| | *nad1*[a] | 885 | ATG | TAA | 295 |
| | *nad2*[a] | 1,461 | ATG | TAA | 486 |
| | *nad3* | 357 | ATG | TAA | 118 |
| | *nad4*[a] | 1,485 | ATG | TGA | 495 |
| NADH dehydrogenase | *nad4L* | 303 | ATG | TAA | 100 |
| | *nad5*[a] | 2,004 | ATG | TAA | 667 |
| | *nad6* | 630 | ATG | TAG | 209 |
| | *nad7*[a] | 1,185 | ATG | TAG | 394 |
| | *nad9* | 573 | ATG | TAA | 190 |
| | *rpl2*[a] | 1,032 | ATG | TAA | 343 |
| Ribosomal proteins (LSU) | *rpl10* | 489 | ATG | TAA | 163 |
| | *rpl16* | 408 | ND | TAA | 136 |
| | *rps3*[a] | 1,641 | ATG | TAA | 547 |
| Ribosomal proteins (SSU) | *rps4* | 969 | ATG | TAA | 322 |
| | *rps7(×2)* | 447 | ATG | TAA | 148 |
| | *rps12* | 390 | ATG | TGA | 129 |
| Succinate dehydrogenase | *sdh4* | 396 | ATG | TAA | 132 |
| Ribosomal RNAs | *rrn5* | 115 | – | – | – |
| | *rrnS* | 1,912 | – | – | – |
| | *rrnL* | 3,321 | – | – | – |
| Transfer RNAs | *trnC-GCA* | 71 | – | – | – |
| | *trnC-ACA* | 71 | – | – | – |
| | *trnD-GUC*[b] | 74 | – | – | – |
| | *trnE-UUC* | 72 | – | – | – |
| | *trnfM-CAU* | 73 | – | – | – |
| | *trnF-UGG* | 75 | – | – | – |

**Table 1** (*continued*)

| Group of genes | Gene name | Length | Start codon | Stop codon | Amino acid |
|---|---|---|---|---|---|
| | *trnG-GCC* | 72 | – | – | – |
| | *trnH-GUG*[b] | 73 | – | – | – |
| | *trnH-AUG* | 70 | – | – | – |
| | *trnK-UUU* | 73 | – | – | – |
| | *trnM-CAU(×3)*[b] | 74 | – | – | – |
| | *trnN-GUU*[b] | 72 | – | – | – |
| | *trnP-GAA* | 74 | – | – | – |
| | *trnP-UGG* | 75 | – | – | – |
| | *trnQ-UUG* | 72 | – | – | – |
| | *trnS-GCU* | 88 | – | – | – |
| | *trnS-UGA* | 87 | – | – | – |
| | *trnS-GGA*[b] | 87 | – | – | – |
| | *trnV-GAC*[b] | 72 | – | – | – |
| | *trnW-CCA*[b] | 74 | – | – | – |
| | *trnY-GUA*[b] | 83 | – | – | – |

**Notes.**
[a]Genes containing introns.
[b]Cp-derived genes.
ND, not determined.

**Table 2  Comparison of genome features in six higher plant mt genomes.**

| Plant species | Coding regions (%) | | | | Non-coding regions (%) |
|---|---|---|---|---|---|
| | Protein-coding genes | *cis*-spliced introns | rRNAs | tRNAs | |
| *G. biloba* | 9.95 | 11.31 | 1.44 | 0.5 | 76.8 |
| *Z. mays* | 6.06 | 4.06 | 0.99 | 0.28 | 88.61 |
| *G. max* | 8.48 | 8.09 | 1.31 | 0.35 | 81.77 |
| *G. raimondii* | 5.14 | 5.28 | 1.61 | 0.28 | 87.69 |
| *P. tremula* | 3.84 | 3.33 | 0.69 | 0.21 | 91.93 |
| *S. suchowensis* | 4.68 | 4.21 | 0.83 | 0.27 | 90.01 |

As shown in Table 2, protein-coding genes and *cis*-introns regions account for 4.68% and 4.21% (27,122 bp) of the whole *S. suchowensis* genome sequence, respectively, while the proportions of rRNA and tRNA regions only represent 0.83% and 0.27%. The remaining regions are non-coding sequences, including introns, intergenic spacers and may be pseudogenes. Specifically, the total length of all the 33 protein-coding genes is 30,141 bp and these genes comprise 10,047 codons. Among these codons, 11.4% (1,141) of them encode *Leucine*, and 1.6% (165) of which encode *Tryptophan*, which are the most and least common amino acids, independently. What's more, with the comparison of genome features in six higher plant mt genomes (Table 2), we found that the coding genes were conserved but the proportions of coding regions were extremely different from each other probably due to their different sizes of mt genomes.

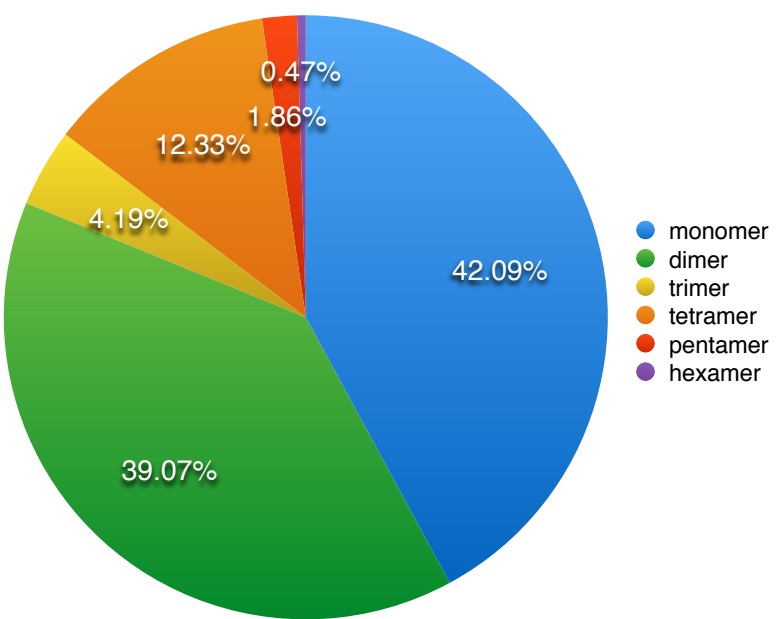

**Figure 2 The distribution of SSRs in the *S. suchowensis* mt genome.** The colors represent different types of SSRs. The percentages of SSRs are also provided on the pie chart.

## Identification of repeat sequences in the *S. suchowensis* mt genome

As described by *Bi et al. (2016)*, repeat sequences are composed of simple sequence repeats (SSRs), tandem repeats, short repeats and large repeats. The simple sequence repeats are a group of tandem repeated sequences with the size of one to six nucleotide repeat units (*Chen et al., 2006*). A total of 430 SSRs were discovered in *S. suchowensis* mt genome, including 181 (42.09%) monomers, 168 (39.07%) dimers, 18 (4.19%) trimers 53 (12.33%) tetramers 8 (1.86%) pentamers and 2 (0.47%) hexamers (Fig. 2). Among the 430 SSRs, more than 81% repeats were belonged to monomers and dimers, which was nearly the same proportion (80%) with that of *Gossypium raimondii* mt genome. Further analysis of the repeat unit of SSRs showed that 88.5% of monomers were A/T contents, whereas G/C only occupied 11.5%. The higher AT contents in mononucleotide SSRs of *S. suchowensis* mt genome was in conformity with the high AT contents (55.02%) in the complete *S. suchowensis* mt genome. The specific size and location of hexamers and pentamers were listed in Table 3, which showed that only one pentamer was located in a protein-coding gene (matR), one located in *cis*-intron (*rps3*), and the others were all located in the intergenic spacers. It is clearly that SSRs distributed in the protein-coding regions are much less than non-coding areas, which account for 3.5% and 96.5%, separately. In non-coding areas, 8.8% of SSRs exists in the introns and 87.7% of SSRs exists in the intergenic spacers.

As illustrated in Table 4, we also identified a total of ten perfect tandem repeats with lengths ranging from 12 bp to 28 bp in the *S. suchowensis* mt genome. All of the ten tandem repeats were discovered in the intergenic spacers. Two long tandem repeats (28 bp) were located in the *nad2/rpl10* and *trnP/rps4* intergenic spacers, three in the same intergenic spacer of *ccmFn/atp8*, and the other five tandem repeats were distributed in the intergenic

**Table 3  Distribution of penta and hexa SSRs in the *S. suchowenisis* mt genome.**

| No. | Type | SSR | Start | End | Location |
|---|---|---|---|---|---|
| 1 | penta | (CCTAA) × 3 | 156,641 | 156,655 | IGS(*trnK-UUU, ccmFc*) |
| 2 | penta | (TATGG) × 3 | 178,046 | 178,060 | IGS(*ccmFc, cox1*) |
| 3 | penta | (GTAAA) × 3 | 298,400 | 298,414 | IGS(*nad6, trnfM-CAU*) |
| 4 | penta | (CCTAA) × 3 | 330,686 | 330,700 | rps3(intron) |
| 5 | penta | (ATAAG) × 3 | 400,749 | 400,763 | IGS(*trnE-UUC, trnQ-UUG*) |
| 6 | penta | (CTAGT) × 3 | 476,584 | 476,598 | *matR* |
| 7 | penta | (CTCTT) × 3 | 527,486 | 527,500 | IGS(*trnS-UGU, nad4*) |
| 8 | penta | (TTTTC) × 3 | 528,072 | 528,086 | IGS(*trnS-UGU, nad4*) |
| 9 | hexa | (ATAAGA) × 3 | 111,445 | 111,462 | IGS(*ccmFn, atp8*) |
| 10 | hexa | (TCCATA) × 3 | 251,465 | 251,482 | IGS(*trnH, ccmB*) |

**Notes.**
IGS, intergenic spacers.

**Table 4  Distribution of perfect tandem repeats in *S. suchowensis* mitogenome.**

| No. | Size(bp) | Start | Stop | Repeat sequence | Location |
|---|---|---|---|---|---|
| 1 | 28 | 6,098 | 6,153 | TATCTATTAATATCTTTTCTTATAATGT (×2) | IGS(*nad2, rpl10*) |
| 2 | 14 | 45,814 | 45,841 | AATATAGAATATAA (×2) | IGS(*trnM-CAU, nad7*) |
| 3 | 13 | 94,082 | 94,107 | TTAGTTTATGAAT (×2) | IGS(*trnV-GAC, ccmFn*) |
| 4 | 12 | 115,524 | 115,547 | GCTTTTGTCAAG (×2) | IGS(*ccmFn, atp8*) |
| 5 | 14 | 124,175 | 124,202 | CTATAAAGATAAAG (×2) | IGS(*ccmFn, atp8*) |
| 6 | 15 | 137,256 | 137,285 | CTTTTATTTTACTTA (×2) | IGS(*ccmFn, atp8*) |
| 7 | 13 | 144,479 | 144,504 | AAGAATGAATTAC (×2) | IGS(*atp8, trnK-UUU*) |
| 8 | 13 | 166,970 | 166,995 | CTCGTATTCTGTA (×2) | IGS(*trnK-UUU, ccmFc*) |
| 9 | 14 | 207,285 | 207,312 | ATCTATCCTACCTA (×2) | IGS(*trnY-GUA, trnN-GUU*) |
| 10 | 28 | 597,277 | 597,332 | TAGGACAGATGTACAAGGTCTTTCTTTA (×2) | IGS(*trnP-UGG, rps4*) |

**Notes.**
IGS, intergenic spacers.

spacers of *trnM-CAU/nad7*, *trnV-GAC/ccmFn*, *atp8/trnK-UUU*, *trnK-UUU/ccmFc* and *trnY-GUA/trnN-GUU*, respectively. The results showed that SSRs and tandem repeats were principally concentrated in the intergenic spacer.

Apart from SSRs and tandem repeats, a total of 349 short and large repeats with the total length of 29,627 bp (4.6% of the mt genome) were detected in the *S. suchowensis* mt genome. The short repeats represent less than 1 kb, while the large repeats are longer than 1 kb. As shown in Fig. 3, most of the repeats (164 repeats, 47%) were ranged from 20 bp to 29 bp, 23.2% were 30 bp to 39 bp long, and fifteen repeats were longer than 100 bp (Table 5), with only one longer than 1 kb (R1:15592 bp). Among the fifteen repeats (>100 bp), six of them were direct repeats, and nine were inverted repeats. Larger repeats are remarkable because they are associated with the changes of mitochondrial reversible genomic structure. Additionally, genes appearing in large repeats may generate multiple copies. In the *S. suchowensis* mt genome, we discovered only one gene of *rps7* in R1 has an intact copy.

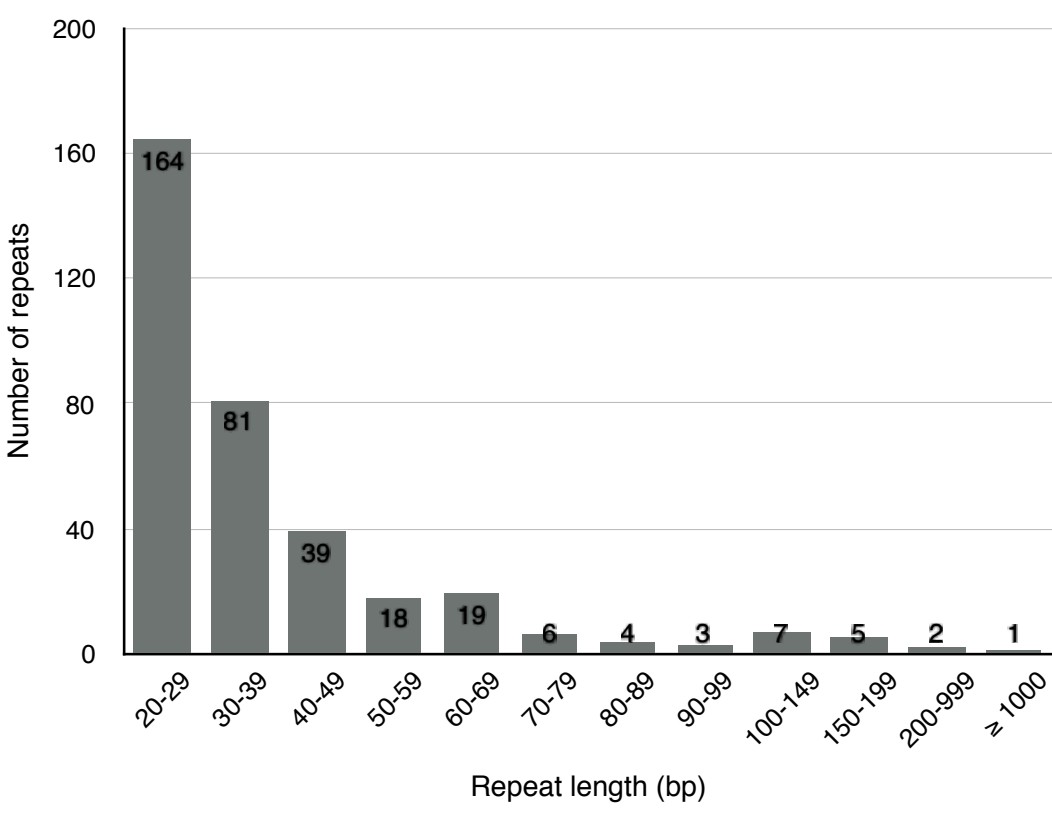

**Figure 3** **The frequency distribution of repeat lengths in the *S. suchowensis* mt genome.** The number shown in histogram represents the specific frequency of each repeat length.

## RNA editing sites in the *S. suchowensis* mt genome

Previous studies have identified that RNA editing is a posttranscriptional process, which converts specific Cytidines to Uridines in chloroplast and mitochondrial genomes of land plants (*Bock & Khan, 2004*; *Chen et al., 2011*; *Raman & Park, 2015*; *Wakasugi et al., 1996*; *Zandueta-Criado & Bock, 2004*). This RNA-editing process is prone to increase the protein conservation among higher plants by "correcting" codons. Exploiting this principle, the PREP-mt program was used to predict 330 RNA editing sites in 33 protein-coding genes (including one multicopy gene: rps7) of the *S. suchowensis* mt genome and 100% C to U RNA editing. As illustrated in Fig. 4, nad4 encoded the most RNA-editing sites (37 sites), while up to five genes (*atp8*, *atp9*, *cox1*, *cox2* and *nad3*) encoded none of RNA editing sites. Among these RNA editing sites, 36.1% (119 sites) occurred in the first base position of the codon, while 63.9% (211 sites) occurred in the second base position and none were in the third position. The occurrence of RNA editing would give rise to the diversity of start and stop codon in protein-coding genes. Additionally, further study found that 46.7% (154 sites) of the RNA editing amino acids were converted from hydrophilic to hydrophobic, 30% (99 sites) from hydrophobic to hydrophobic, 14% (46 sites) from hydrophilic to hydrophilic, 9.1% (30 sites) from hydrophobic to hydrophilic, and only one amino acids of *ccmFc* gene was converted from Arginine to a stop codon. Among these, most of the amino acids tended to be converted from Serine to Leucine (21.8%, 72 sites), Proline to

**Table 5  Distribution of repeats (>100 bp) in the *S. suchowensis* mt genome.**

| No. | Identity(s) | Copy-1 | | Copy-2 | | Size(bp) | Type[a] |
|---|---|---|---|---|---|---|---|
| | | Start | End | Start | End | | |
| R1 | 99.99 | 540,902 | 556,493 | 644,437 | 628,845 | 15,592 | IR |
| R2 | 99.64 | 324,394 | 324,672 | 420,173 | 420,450 | 279 | DR |
| R3 | 98.85 | 72,667 | 72,926 | 157,110 | 156,851 | 260 | IR |
| R4 | 98.5 | 454,848 | 455,046 | 626,985 | 627,184 | 199 | DR |
| R5 | 100 | 27,411 | 27,609 | 358,568 | 358,370 | 199 | IR |
| R6 | 98.34 | 218,990 | 219,170 | 336,654 | 336,834 | 181 | DR |
| R7 | 98.86 | 54,691 | 54,866 | 344,482 | 344,307 | 176 | IR |
| R8 | 99.39 | 11,483 | 11,646 | 195,066 | 195,229 | 164 | DR |
| R9 | 98.64 | 25,593 | 25,739 | 274,499 | 274,354 | 147 | IR |
| R10 | 99.22 | 384,916 | 385,044 | 594,181 | 594,309 | 129 | DR |
| R11 | 94.69 | 183,152 | 183,264 | 429,752 | 429,864 | 113 | DR |
| R12 | 100 | 602,388 | 602,496 | 619,004 | 618,896 | 109 | IR |
| R13 | 98.08 | 331,484 | 331,587 | 483,041 | 482,939 | 104 | IR |
| R14 | 84.11 | 319,298 | 319,401 | 378,261 | 378,159 | 104 | IR |
| R15 | 92.31 | 433,759 | 433,858 | 604,908 | 604,805 | 100 | IR |

**Notes.**
[a] DR or IR: direct or inverted repeats, respectively.

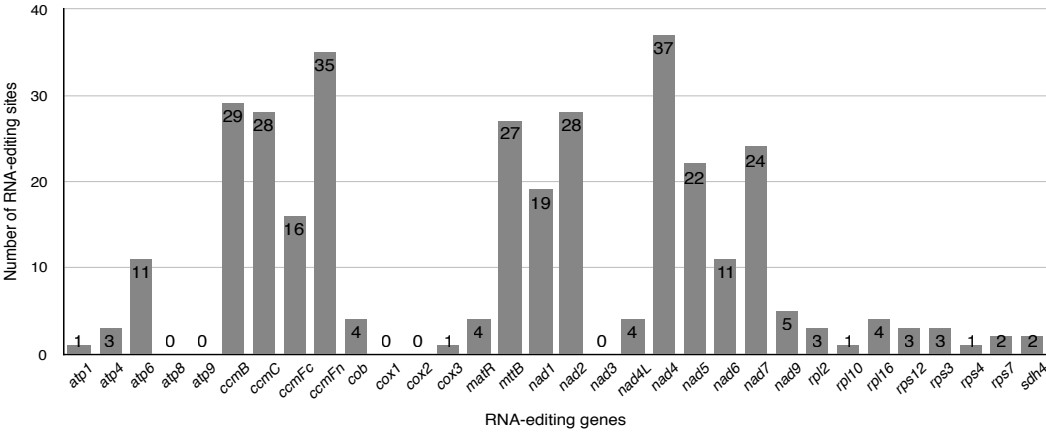

**Figure 4  The distribution of RNA editing sites in the *S. suchowensis* mt protein-coding genes.** The number shown by gray box represents the RNA editing sites of each gene.

Leucine (20.3%, 67 sites) and Serine to Phenylalanine (15.8%, 52 sites). The remaining 139 RNA editing sites are distributed in other RNA editing types, including Ala to Val, His to Tyr, Leu to Phe, Pro to Phe, Pro to Ser, Arg to Trp, Thr to Met, Thr to Ile, and Arg to X (X = stop codon).

## The substitution rates of protein-coding genes in the *S. suchowensis* mt genome

In genetics, nonsynonymous (Ka) and synonymous (Ks) substitution rates is significant for understanding evolutionary dynamics of protein-coding genes across similar and yet

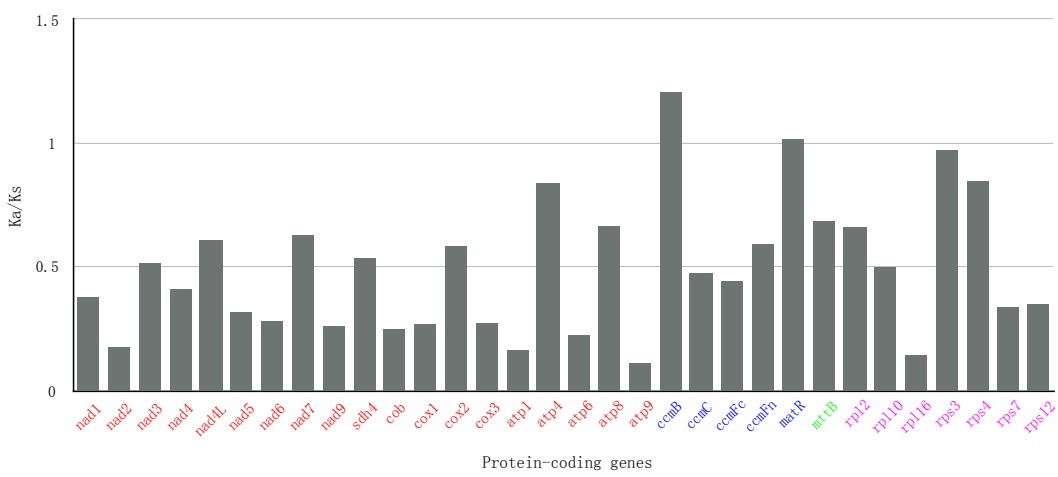

**Figure 5** **The Ka/Ks values of 32 protein-coding genes of *S. suchowensi s* versus *G. raimondii*.** The red, blue, green and purple genes indicate Complex genes, Cytochrome c biogenesis, Transport membrane protein and Ribosomal proteins, respectively.

diverged species. It has been widely known that Ka/Ks indicate neutral evolution when Ka/Ks equals to 1, positive selection when Ka/Ks greater than 1, and negative selection when Ka/Ks less than 1. In this study, all of the 32 protein-coding genes of *S. suchowensis* mt genome were used to calculate the Ka and Ks substitution rates against the *Gossypium raimondii* mt genome. As shown in Fig. 5, the ratios of Ka/Ks were significantly less than 1 in most of the protein-coding genes. Nevertheless, the Ka/Ks ratio of *ccmB* (1.21) was greater than 1, indicating that this gene may have suffered from positive selection since the divergence of *S. suchowensis* and *G. raimondii* from their last common ancestor (*Bi et al., 2016*). Additionally, two genes (*matR*: 1.02, *rps3*: 0.97) were close to 1, indicating that they may have experienced neutral evolution since the divergence of their common ancestor. Combining with Table 1, the Fig. 5 also illustrated that the Ka/Ks ratios of all the Complex genes and Ribosomal proteins were all below 1, as well as a transport membrane protein (*mttB*), indicating that purifying selection was acting on these genes, and these genes may be highly conservative in the evolutionary process of higher plants.

## Migration of chloroplast DNA in mitochondrial genomes

Without the full-length nuclear genome sequence of *S. suchowensis*, we cannot identify the DNA transfer between the nucleus and mitochondria. The *S. suchowensis* mt genome was searched against its available cp genome and the similar sequences were filtered, which resulted in 41 matched hits between the two genomes. The hits covered 11.3% (17.5 kb) of the cp genome and 2.8% (18.1 kb) of the mt genome. Interestingly, when the mt genome was searched against the representative seed plant cp genomes, the matched hits covered 3.2% (20.5 kb) of the mt genome, suggesting that some sequences were lost from the *S. suchowensis* cp genome but were maintained by mt genome after the ancient chloroplast to mitochondrial transfers.

The protein-coding and tRNA genes transferred from cp to mt genomes were also identified by NCBI-BLASTn with stringent parameters. The results showed that only two

complete cp protein-coding genes (*atpE* and *psbD*) and nine tRNA genes (*trnD-GUC, trnH-GUG, trnK-GUG, trnM-CAU, trnN-GUU, trnS-GGA, trnV-GAC, trnW-CCA* and *trnY-GUA*) were transferred to mt genome (Table 1). Additionally, some partial cp protein-coding genes (*accD, atpA, atpB, clpP, petB, psaA, psaB, psbA, psbB, psbC, rpoB, rps12* and *ycf2*) were also transferred to the mt genome. The distinction of transferred genes between protein-coding and tRNA genes suggested that protein-coding genes could not survive completely during the evolution. However, tRNA genes were preserved probably because they played important roles in the *S. suchowensis* mt genome.

## Phylogenetic analysis within higher plant mt genomes

Phylogenetic analyses were performed based on an aligned data matrix of 35 species and 23 protein-coding genes. As illustrated in Fig. 6, the 35 plant species were divided into four categories, including 19 rosids (*Glycine max, Vigna angularis, Millettia pinnata, Lotus japonicus, Medicago truncatula, Vitis vinifera, Malus domestica, Gossypium hirsutum, Gossypium raimondii, Carica papaya, Batis maritima, Brassica napus, Arabidopsis thaliana, Raphanus sativus, Populus tremula, Salix suchowensis, Citrullus lanatus, Cucumis sativus* and *Cucurbita pepo*), 8 asterids (*Daucus carota, Capsicum annuum, Nicotiana tabacum, Beta vulgaris, Ajuga reptans, Salvia miltiorrhiza, Asclepias syriaca* and *Rhazya stricta*), 6 monocots (*Butomus umbellatus, Phoenix dactylifera, Oryza sativa, Zea mays, Sorghum bicolor* and *Tripsacum dactyloides*) and 2 gymnosperms (*Cycas taitungensis* and *Ginkgo biloba*). The two gymnosperms were designated as outgroup. Abbreviations of all these observed plant mt genomes are listed in Table S3. Bootstrap analysis showed that 27 of the 32 nodes were supported by values over 70%, 22 out of 32 nodes with bootstrap values over 90% and 11 nodes supported by 100% bootstrap value (Fig. 6). The phylogenetic tree strongly supported the separation of rosids and asterids with 99% bootstrap value, as well as the separation of eudicots and monocots (100%), and the separation of angiospermae and gymnospermae (100%). Additionally, according to the NJ tree, *S. suchowensis* and *P. tremula* were classified into one clade (Salicaceae) with 100% bootstrap value, whereas this clade was a little different from other plant species in rosids.

## Comparison of genomic features with other mitochondrial genomes

With the rapid development of sequencing technology, more and more complete plant mt genomes have been assembled, which would provide an important opportunity to compare the genomic features within multiple plant species (*Alverson et al., 2011*; *Wei et al., 2016*). As described by *Richardson et al. (2013)*, the plant mt genomes varied considerably in size, gene content and order, only the same genus sharing the similar genomic features. In genus *Salix*, another mt genome of *Salix purpurea* has been assembled. It is very important to make comparative genomic analyses between *S. suchowensis* and *S. purpurea* mt genome. As illustrated in Fig. 7, the size (598,970 bp) and GC content (44.94%) of *S. purpurea* are a little less than that of *S. suchowensis* mt genome (*Alverson et al., 2011*; *Wei et al., 2016*). What's more, the number of genes and introns in the two mt genomes are similar, with *S. suchowensis* having only one tRNA genes (*trnH-AUG*) annotated than that of *S. purpurea*. By comparing the genome physical map of *S. purpurea* and *S. suchowensis* mt genome, we

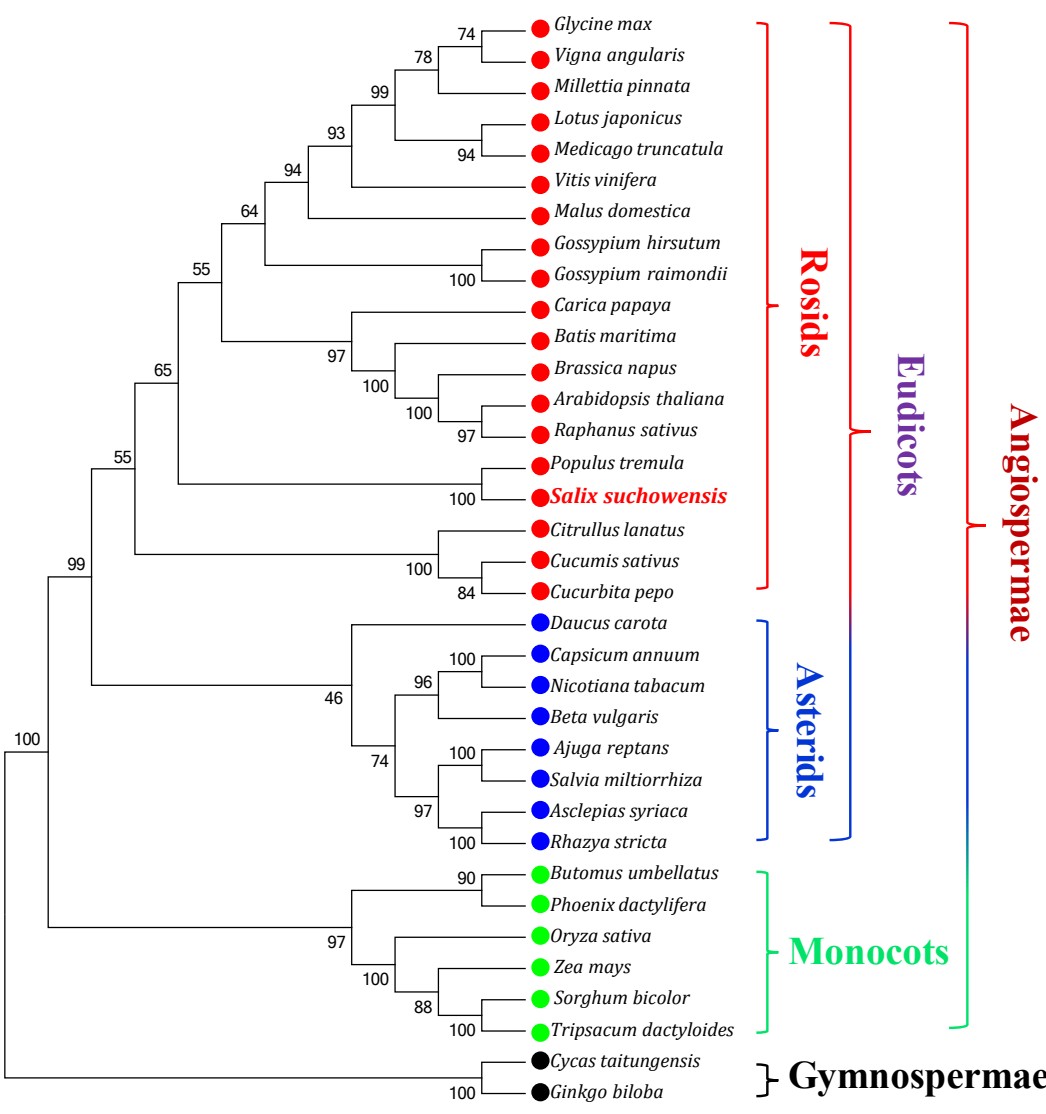

**Figure 6** **The Neighbor-Joining tree was constructed based on 23 conserved genes of 35 representative plant mt genomes.** The colors of red, blue, green and black circles represent the classes of rosids, asterids, monocots and gymnosperms. Numbers at the nodes are bootstrap support values.

found that gene orders of the two mt genomes were different from each other, whereas only a few gene orders were conserved during evolution. The conserved gene orders of the two mt genomes are as follows: *nad7-trnV* (4 genes), *ccmFn-trnK* (3 genes), *trnY-ccmB* (7 genes), *rpl16-matR* (13 genes), *nad3-rps7* (5 genes), *atp6-trnS* (3 genes) and *trnP-rps7* (4 genes). From Table S4, we found that nearly all of the *S. suchowensis* mt genome sequence showed a high similarity (most identity > 99%) with that of *S. purpurea* mt genome, and the longest syntenic blocks was 143,567 bp. The extremely high identities between *S. suchowensis* and *S. purpurea* mt genome suggest that the differentiation of the two *Salix* species must be very recent during the evolution of plants.

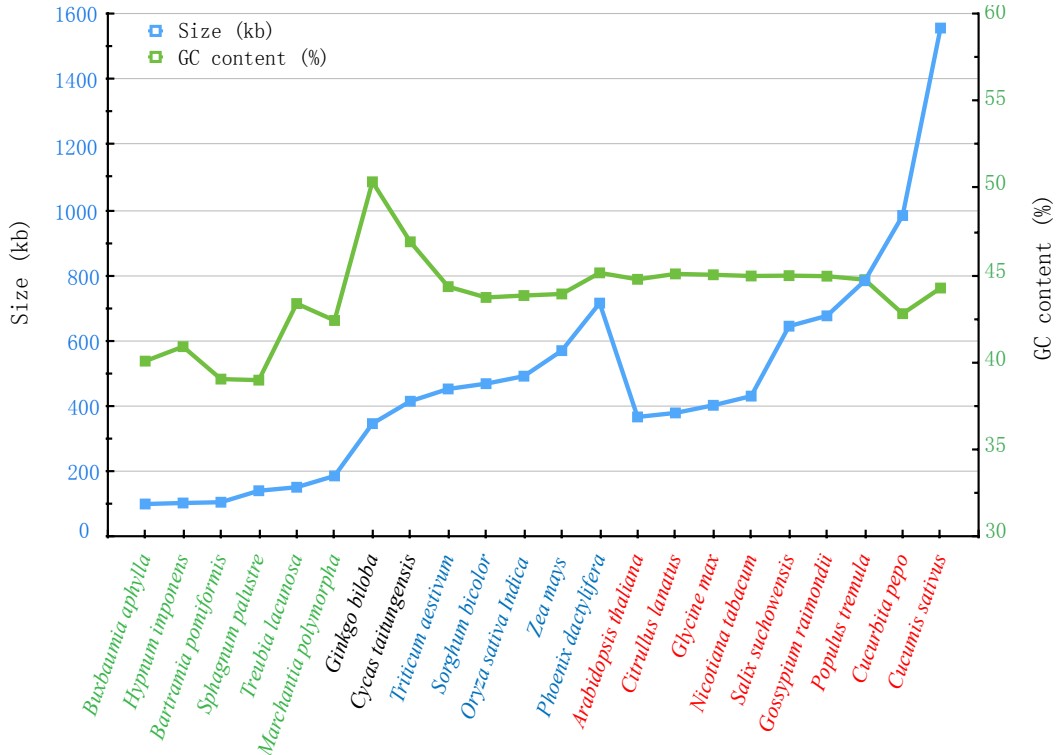

**Figure 7** **Comparison of sizes and GC contents within 22 land plant mt genomes.** The blue and green broken lines represent the sizes and GC contents, respectively. The colors of green, black, blue and red indicate different species in Bryophyte, Gymnospermae, Monocotyledoneae and Dicotledonae, respectively.

Additionally, we also compared the sizes and GC contents of 22 land plant mt genomes (Fig. 7), including 5 bryophytes (*Buxbaumia aphylla*, *Bartramia pomiformis*, *Sphagnum palustre*, *Treubia lacunosa* and *Marchantia polymorpha*), 2 gymnosperms (*Ginkgo biloba* and *Cycas taitungensis*), 5 monocots (*Triticum aestivum*, *Sorghum bicolor*, *Oryza sativa*, *Zea mays* and *Phoenix dactylifera*) and 8 dicots (*Arabidopsis thaliana*, *Citrullus lanatus*, *Glycine max*, *Nicotiana tabacum*, *Salix suchowensis*, *Salix purpurea*, *Gossypium raimondii*, *Populus tremula*, *Cucurbita pepo* and *Cucumis sativus*). The abbreviation of all these observed plant mt genomes are listed in Table S3. The sizes of our observed plant mt genomes vary from 100,725 bp in *B. aphylla* to 1,555,935 bp in *C. sativus*, while the size of *S. suchowensis* (644,437 bp) mt genome is in the middle of these observed plants. Similarly, the GC contents are also variable, ranging from 38.99% in *S. palustre* to 50.36% in *G. biloba*. Additionally, the GC contents of angiosperms, including monocots and dicots, are larger than that of bryophytes but smaller than that of gymnosperms, suggesting that the GC contents changed frequently since the divergence of bryophytes, gymnosperms and angiosperms. However, the GC contents in angiosperms were conserved during the evolution, despite their genome sizes varied tremendously.

In the *S. suchowensis* mt genome, a total of 23 introns, including 17 *cis*-introns and 6 *trans*-introns, were discovered in nine genes. Among the nine intron-containing genes, three of which (*nad1*, *nad2* and *nad5*) require *cis*- splicing as well as *trans*- splicing events

**Table 6  Comparison of *cis-/trans-* splicing introns in four plant mt genomes.**

| Gene | G. biloba | T. aestivum | P. tremula | S. suchowensis |
|------|-----------|-------------|------------|----------------|
| nad1 | 2/2 | 1/3 | 2/2 | 2/2 |
| nad2 | 3/1 | 3/1 | 3/1 | 3/1 |
| nad4 | 3/– | 3/– | 3/– | 3/– |
| nad5 | 2/2 | 2/2 | 2/2 | 2/2 |
| nad7 | 4/– | 4/– | 4/– | 4/– |
| atp1 | – | – | – | –/1 |
| ccmFc | 1/– | 1/– | 1/– | 1/– |
| cox2 | 2/– | 1/– | – | – |
| rpl2 | 1/– | – | 1/– | 1/– |
| rps3 | 2/– | 3/– | 1/– | 1/– |
| rps10 | – | 1/– | – | – |
| Total | 20/5 | 19/6 | 17/5 | 17/6 |

to maturate, while *atp1* only requires *trans*-splicing event and the other five genes (*nad4*, *nad7*, *rpl2*, *ccmFc* and *rps3*) only require *cis*-splicing event to maturate (Table 6). During the evolution of seed plant mt genomes, genes containing introns were conserved, as well as the numbers of *cis*- or *trans*- introns. Table 6 compared the number of *cis*- and *trans*- introns in four seed plant mt genomes, including one gymnosperm (*G. biloba*), one monocot (*T. aestivum*), and two dicots (*P. tremula* and *S. suchowensis*). As illustrated in Table 6, *G. biloba* contains 20 *cis*- and 5 *trans*- introns, *T. aestivum* contains 19 *cis*- and 6 *trans*- introns, and *P. tremula* contains 17 *cis*- and 5 *trans*- introns. The numbers of introns in *nad2*, *nad4*, *nad5* and *nad7* are extremely conserved, but there are still some introns lost or maintained during the evolution. For example, a *trans*-intron in *atp1* was lost from *G. biloba*, *T. aestivum*, *P. tremula* but maintained by *S. suchowensis*; a *cis*-intron in *rps10* was maintained by *T. aestivum* but lost from the others. Additionally, one or two *cis*-introns in *cox2* were lost from Salicaceae during the evolution. Overall, the locations and numbers of introns in the *S. suchowensis* mt genome were rather well conserved as in other complete seed plant mt genomes.

Previous studies revealed that the length of homologous sequences among different plant mt genomes was consistent with taxonomies, despite the unique variability among these mt genomes (*Li et al., 2009b*). The closely related species always share the greatest sequences, even in the non-coding regions, while species belong to different families share fewer (*Li et al., 2009b*). During the evolution of plant mt genomes, the loss and acquisition of protein-coding and tRNA genes occurred frequently. Table 7 compared the protein-coding and tRNA genes of five representative plant mt genomes. According to Tables 1 and 7, we found that the numbers of protein-coding genes in mt genomes tend to decrease during the evolution of plants. In these observed plant mt genomes, most of the protein-coding genes were conserved in different plant mt genomes, however, the ribosomal proteins were highly variable. The *rpl5* gene was only lost in Salicaceae but existed in other observed plant mt genomes, while the *rpl10* gene was only found in Salicaceae but lost in others. Three ribosomal proteins (*rps2*, *rps13* and *rps19*) seemed to have been lost in eudicots after the

**Table 7  Comparison of genome features of five plant mt genomes.**

| Gene | Brassicales | Salicaceae | | Poales | Ginkgoales | Marchantiaceae |
|---|---|---|---|---|---|---|
| | *Arabidopsis thaliana* | *Populus tremula* | *Salix suchowensis* | *Oryza sativa* | *Ginkgo biloba* | *Marchantia polymorpha* |
| Size(bp) | 366,924 | 783,442 | 644,437 | 491,515 | 346,544 | 186,609 |
| GC(%) | 44.77 | 44.75 | 44.98 | 43.84 | 50.36 | 42.41 |
| *nad1* | + | + | + | + | + | + |
| *nad2* | + | + | + | + | + | + |
| *nad3* | + | + | + | + | + | + |
| *nad4* | + | + | + | + | + | + |
| *nad4L* | + | + | + | + | + | + |
| *nad5* | + | + | + | + | + | + |
| *nad6* | + | + | + | + | + | + |
| *nad7* | + | + | + | + | + | pseudo |
| *nad9* | + | + | + | + | + | + |
| *sdh3* | − | − | − | − | + | + |
| *sdh4* | − | + | + | − | + | + |
| *cob* | + | + | + | + | + | + |
| *cox1* | + | + | + | + | + | + |
| *cox2* | + | + | + | + | + | + |
| *cox3* | + | + | + | + | + | + |
| *atp1* | + | + | + | + | + | + |
| *atp4* | + | + | + | + | + | + |
| *atp6* | + | + | + | + | + | + |
| *atp8* | + | + | + | + | + | + |
| *atp9* | + | + | + | + | + | + |
| *ccmB* | + | + | + | + | + | + |
| *ccmC* | + | + | + | + | + | + |
| *ccmFC* | + | + | + | + | + | + |
| *ccmFN* | + | + | + | + | + | + |
| *matR* | + | + | + | + | + | + |
| *mttB* | + | + | + | + | + | + |
| *rpl2* | + | + | + | + | + | + |
| *rpl5* | + | − | − | + | + | + |
| *rpl6* | − | − | − | − | − | + |
| *rpl10* | − | + | + | − | − | − |
| *rpl16* | + | pseudo | + | + | + | + |
| *rps1* | − | + | − | + | + | + |
| *rps2* | − | − | − | + | + | + |
| *rps3* | + | + | + | + | + | + |
| *rps4* | + | + | + | + | + | + |
| *rps7* | + | + | + | + | + | + |
| *rps8* | − | − | − | − | − | + |

**Table 7** (*continued*)

| Gene | Brassicales | Salicaceae | | Poales | Ginkgoales | Marchantiaceae |
| --- | --- | --- | --- | --- | --- | --- |
| | *Arabidopsis thaliana* | *Populus tremula* | *Salix suchowensis* | *Oryza sativa* | *Ginkgo biloba* | *Marchantia polymorpha* |
| *rps10* | + | − | − | − | + | + |
| *rps11* | − | − | − | pseudo | + | + |
| *rps12* | + | + | + | + | + | + |
| *rps13* | − | − | − | + | + | + |
| *rps14* | + | + | − | pseudo | + | + |
| *rps19* | − | − | − | + | + | + |
| *trnA-UGC* | − | − | − | − | − | + |
| *trnC-GCA* | + | + | + | + | + | + |
| *trnD-GUC* | + | + | + | + | + | + |
| *trnE-UUC* | + | + | + | + | + | + |
| *trnF-GAA* | − | + | + | + | + | + |
| *trnG-GCC* | + | + | + | − | + | + |
| *trnG-UCC* | − | − | − | − | − | + |
| *trnH-GUG* | + | + | + | + | + | + |
| *trnI-CAU* | + | − | − | + | − | + |
| *trnI-GAU* | − | − | − | − | − | − |
| *trnK-UUU* | + | + | + | + | + | + |
| *trnL-CAA* | − | − | − | − | − | + |
| *trnL-UAA* | − | − | − | − | + | + |
| *trnL-UAG* | − | − | − | − | + | + |
| *trnM-CAU* | + | + | + | + | + | + |
| *trnfM-CAU* | + | + | + | + | + | + |
| *trnN-GUU* | + | + | + | + | − | + |
| *trnP-UGG* | + | + | + | + | + | + |
| *trnQ-UUG* | + | + | + | + | + | + |
| *trnR-ACG* | − | − | − | − | + | + |
| *trnR-UCG* | − | − | − | − | − | + |
| *trnR-UCU* | − | − | − | − | + | + |
| *trnS-GCU* | + | + | + | + | + | + |
| *trnS-UGA* | + | + | + | + | + | + |
| *trnS-GGA* | + | + | + | + | − | − |
| *trnT-GGU* | − | − | − | − | − | + |
| *trnT-UGU* | − | − | − | − | − | − |
| *trnV-UAC* | − | − | − | − | − | + |
| *trnV-GAC* | − | − | + | − | − | − |
| *trnW-CCA* | + | + | + | + | + | + |
| *trnY-GUA* | + | + | + | + | + | + |

differentiation of eudicots and monocots. The *rpl6* and *rps8* genes probably, only existed in *M. polymorpha*, were lost in angiosperms and gymnosperms during the evolution. Strikingly, some functional genes changed into pseudo genes during the evolution, such as *rpl16* in *P. tremula*, *nad7* in *M. polymorpha*, as well as *rps11* and *rps14* in *O. sativa*. Previous studies have shown that the corresponding mitochondrial *rps2* and *rps11* genes have been transferred to the nucleus in *Arabidopsis*, soybean and tomato, suggesting that gene loss events followed by functional transfer to the nucleus(*Palmer et al., 2000*; *Perrotta, Grienenberger & Gualberto, 2002*). Most ribosome protein genes are frequently absent in clades of angiosperm mt genomes, which can be considered unnecessary to some extent (*Shengxin et al., 2013*). Apart from ribosomal proteins, succinate dehydrogenase genes were usually lost in angiosperms. A previous study by Southern blot survey of multiple angiosperms showed that the losses of *sdh4* genes were focused on the monocots, and no losses were detected in basal angiosperms (*Adams et al., 2001*). More specially, contrasting with *G. biloba* and *M. polymorpha*, plant species in angiosperms have lost *sdh3* genes, which can be inferred as *sdh3* gradually lost or developed into pseudo genes during the evolution of angiosperms. Previous studies proposed that genes losses rarely occurred with respiratory genes and most genes occurred with ribosomal proteins (*Shengxin et al., 2013*), which were given support by our data.

The protein synthesis of plant mt genomes requires an entire set of tRNA genes, including *trnA*, *trnC*, *trnD*, *trnE*, *trnF*, *trnG*, *trnH*, *trnI*, *trnK*, *trnL*, *trnM*, *trnN*, *trnP*, *trnQ*, *trnR*, *trnS*, *trnT*, *trnV*, *trnW*, *trnY* and *trnfM*. However, a large number of tRNA genes experienced inactivation, loss or migration during the evolution of plant mt genomes (*Dietrich et al., 1996*). As shown in Table 7, six tRNA genes (*trnA-UGC*, *trnG-UCC*, *trnL-CAA*, *trnR-UCG*, *trnT-GGU* and *trnV-UAC*) were lost in all observed plants except *M. polymorpha*, suggesting that this loss event probably occurred in the differentiation of bryophytes and gymnosperms. Additionally, four tRNA genes (*trnL-UAG*, *trnL-UAA*, *trnR-ACG* and *trnR-UCU*) were lost from gymnosperms to angiosperms. Specially, five kinds of tRNA genes (*trnA*, *trnI*, *trnL*, *trnR* and *trnT*) were lost from the *S. suchowensis* mt genome, suggesting that tRNA genes importing from cytosol must be invoked as the mechanism for making up the deficit (*Kubo et al., 2000*). Identification of the number and types of tRNA genes in *S. suchowensis* mt genome will be helpful to evaluate the origin and evolution of tRNA genes in higher plants. Consequently, the above results powerfully suggest that intracellular genes of ribosomal protein and tRNA genes transfer from mitochondria to the nuclear genome are a frequent process. In return, this can also allow the nucleus to control the organelle by encoding organelle-destined proteins (*Woodson & Joanne, 2008*).

In summary, based on the comparison of protein-coding and tRNA genes with other mt genomes, the *S. suchowensis* mt genome has lost 14 genes, including 9 protein-coding genes (*rpl5, rps1, rps2, rps10, rps11, rps13, rps14, rps19* and *sdh3*), 4 tRNA genes (*trnL-UAA, trnL-UAG, trnR-ACG* and *trnR-UCU*), when compared with the representative gymnosperm *G. biloba*. The 9 protein-coding genes, except *rps1*, also can not be found in another Salicaceae mt genome *P. tremula*. Previous studies in other plant mt genomes showed that a few lost mt genes were found to have transferred into the nuclear genomes (*Adams et al., 2000*; *Adams et al., 2002*; *Adams et al., 2001*). Thus, the genes lost may be considered as redundant

genes for *S. suchowensis* mt genome or their functions can be duplicated by other genes, which is an evolutionary compaction of the mt genome in seed plants.

## CONCLUSIONS

In this study, the complete mt genome of *S. suchowensis* shares most of the common genomic features with other plant mt genomes. The identification of RNA editing sites will provide important clues for predicting gene functions with novel codons. The Ka/Ks analysis and the comparison of genomic features with other plant mt genomes should be contribute to a comprehensive understanding of plant mt evolution. Mitochondria are very important for plant breeding. With the accomplishment of the mt genome of *S. suchowensis* and further development of the next and third generation sequencing platforms, we also have an opportunity to conduct further genomic breeding studies in willow.

## ACKNOWLEDGEMENTS

The authors are deeply grateful to Changwei Bi and Xuelin Wang, who performed the experiment and provided valuable suggestions to this manuscript. The authors thank Juan Li and Dongyang Wu for their contributions in annotating and analyzing the mitochondrial genome, and we also thank Yiqing Xu and Qiaolin Ye for drawing some figures.

### Funding

This study was supported by the National Key Research and Development Plan of China (2016YFD0600101), the Fundamental Research Funds for the Central Non-Profit Research Institution of CAF (CAFYBB2014QB015), the National Natural Science Foundation of China (31570662, 31500533, and 61401214), Jiangsu Provincial Department of Housing and Urban-Rural Development (2016ZD44), and the PAPD (Priority Academic Program Development) program at Nanjing Forestry University. The funders had no role in study design, data collection and analysis, decision to publish, or preparation of the manuscript.

### Grant Disclosures

The following grant information was disclosed by the authors:
National Key Research and Development Plan of China: 2016YFD0600101.
Fundamental Research Funds: CAFYBB2014QB015.
National Natural Science Foundation of China: 31570662, 31500533, 61401214.
Jiangsu Provincial Department of Housing and Urban-Rural Development: 2016ZD44.
Nanjing Forestry University.

### Competing Interests

The authors declare there are no competing interests.

## Author Contributions

- Ning Ye conceived and designed the experiments, wrote the paper, prepared figures and/or tables.
- Xuelin Wang performed the experiments, reviewed drafts of the paper.
- Juan Li analyzed the data.
- Changwei Bi analyzed the data, wrote the paper, prepared figures and/or tables.
- Yiqing Xu and Qiaolin Ye contributed reagents/materials/analysis tools.
- Dongyang Wu performed the experiments.

## Data Availability

The mitochondrial genome of *Salix suchowensis* described here are accessible via GenBank accession number NC_029317.1 (https://www.ncbi.nlm.nih.gov/nuccore/NC_029317.1).

## Supplemental Information

Supplemental information for this article can be found online at http://dx.doi.org/10.7717/peerj.3148#supplemental-information.

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
