# Peer review of "Assembly and comparative analysis of complete mitochondrial genome sequence of an economic plant Salix suchowensis"

_PeerJ, doi:10.7717/peerj.3148_

## Round 0.1 · original submission · Major Revisions

· Academic Editor

Major Revisions

Please address comments about comparison with a congeneric species and a possibility of misassembly. Other comments are mainly editorial.

Reviewer 1 ·

Basic reporting

Some correction in presentation of plant names required (see attached file).

Experimental design

No comment.

Validity of the findings

No comment.

Additional comments

As a botanist, I read this paper with great interest, and I found some important suggestions for me. The manuscript is well written and logically consistent. I suggest only few corrections; the most of them concern the writing of plant names (see attached file).

Annotated reviews are not available for download in order to protect the identity of reviewers who chose to remain anonymous.

Reviewer 2 ·

Basic reporting

Overall the article is clearly written and easy to follow. There are however some cases of unclear wording, excessively long sentences. Some relevant references are missing. The most critical is that the authors seem to be unaware about the publication of mitochondrial genome of a congeneric species Salix purpurea (10.1186/s40064-016-3521-6).

Experimental design

The methods are reasonable though there are few points where they require revision.
1) I'm puzzled about the method of sequencing. The authors write (lane 86-88) that they used two technologies, 454 and Illumina, but all description of assembly process concerns 454 data only. How were Illumina data used?
2) As soon as Salix purpurea mt genome is available, please make comparative analysis (GC-content, gene and intron content, gene order, sequence divergence).
3) The presence of chloroplast-derived regions in mitochondrial genomes is a potential for misassembly and emergence of chimeric contigs/scaffolds that contain fragments of mitochondrial and chloroplast genomes. Such misassemblies can be found by mapping of reads back on the assembly and inspecting the coverage (fragments from cp genome have higher coverage). Please check whether the assembly does not have the regions with anomalous coverage.

Validity of the findings

no comment

Additional comments

lane 41 - exceptional stems? unclear, please explain
lane 54-58 - unclear, please rephrase
lane 67-69 - the largest mt genomes are among Cucurbitaceae (10.1186/1471-2164-12-424) and Caryophyllaceae (:10.1371/journal.pbio.1001241), not Corchorus capsularis. The smallest is that of Viscum scurruloideum (10.1073/pnas.1504491112)
lane 74 - Populus trichocarpa - should be in italics
lane 89 - change Hisep to HiSeq
lane 142-143 - not exactly so. There are few examples of DNA transfer from mt to cp genome (see 10.1111/nph.12704). Please remove or revise this sentence
lane 343-344 - not striking, cox2 introns are lost in many plants (e.g. 10.1086/319586, 10.1007/s00438-002-0657-6)

---

## Round 0.2 · Minor Revisions

· Academic Editor

Minor Revisions

While most reviewers' concerns have been addressed, there are a number of points that I do not understand.

(1) What does it mean that "contigs are essential for assembly" - how does that prove that these contigs are not contamination?

(2) There seems to be something wrong with the wording in "these three contigs are essential for assembling the complete mt genome, suggesting that these three contigs may be derived from its cp genome" - to what genome does the pronoun "its" refer?

(3) I suspect that "contigs assembled twice" may be an indication of a problem with the assembly. If not, this needs to be clarified.

---

## Round 0.3 · accepted · Accept

· Academic Editor

Accept

I hope that the readers will understand these sentences as intended by the authors.